# It Takes a Village: Multidisciplinary Approach to Screening and Prevention of Pediatric Sleep Issues

**DOI:** 10.3390/medsci6030077

**Published:** 2018-09-14

**Authors:** Jessica R. Sevecke, Tawnya J. Meadows

**Affiliations:** Department of Psychiatry, Geisinger Health System, 100 North Academy Avenue, Danville, PA 17821, USA; tjmeadows@geisinger.edu

**Keywords:** multidisciplinary, pediatric sleep, anticipatory guidance

## Abstract

Sleep is essential to human development. Poor sleep can have significant effects on cognition, learning and memory, physical and behavioral health, and social-emotional well-being. This paper highlights the prevalence of common pediatric sleep problems and posits that a multidisciplinary approach to the assessment and intervention of sleep problems is ideal. Primary care providers are often the first professionals to discuss sleep issues with youth and families. However, dentists, otolaryngologists, childcare providers, school personnel, and behavioral health providers have a vital role in screening and prevention, providing intervention, and monitoring the progress of daily functioning. The strengths of this approach include better provider-to-provider and provider-to-family communication, streamlined assessment and intervention, earlier identification of sleep issues with more efficient referral, and longer-term monitoring of progress and impact on daily functioning. Barriers to this approach include difficulty initiating and maintaining collaboration among providers, limited provider time to obtain the necessary patient permission to collaborate among all multidisciplinary providers, lack of financial support for consultation and collaboration outside of seeing patients face-to-face, geographic location, and limited resources within communities. Research investigating the utility of this model and the overall impact on pediatric patient sleep issues is warranted and strongly encouraged.

## 1. Introduction

Sleep is essential to human development across all life stages. Adequate sleep is important for physical growth, cognitive development, learning and memory, physical health, social-emotional well-being, and mental and behavioral health. If an individual obtains inadequate sleep, significant implications across these domains could occur [1,2,3,4]. Among adult US citizens, about 50 to 70 million report experiencing a sleep disorder [5]. Because sleep issues often present in childhood and may continue to have impact into adulthood, early identification of medical sleep disorders and behaviorally based sleep problems is essential to prevent future problems such as persistent insomnia or death secondary to issues such as driving while drowsy [5]. Among children, sleep is crucial to ensure that developmental milestones are met and that children thrive academically, socially, and behaviorally across the systems and environments in which they function. The prevalence of sleep issues among infants, children, and adolescents varies according to type of sleep concern and age. Medically based sleep disorders include diagnoses such as obstructive sleep apnea (OSA), bruxism, restless leg syndrome (RLS), and periodic limb movement disorder (PLMD). Medically based sleep disorders are less common than behaviorally based sleep problems, with about 5% of children and adolescents experiencing a medically-based sleep disorder [6] and up to about 40% of children experiencing a behaviorally based sleep problem sometime during childhood [6]. Behaviorally based sleep problems include night wakings, bedtime issues, inadequate sleep, and poor sleep hygiene. In one study, about 20% of 4- to 12-year-old children experienced a behaviorally based sleep problem at least one night per week [7]. Children who rated higher on parasomnias were also more likely to experience falls and exhibit pica symptoms [7]. Despite this level of prevalence, less than half of the parents in this study discussed sleep issues with their primary care provider [7]. Another study suggests that low prevalence rates of sleep issues (3.7%) within primary care compared to epidemiological studies may be due to primary care providers not asking families about sleep problems [6].

## 2. Pediatric Sleep Across Disciplines

### 2.1. Primary Care Providers

Commonly, a child’s primary care provider is the first person to discuss sleep issues and short- and long-term implications of sleep problems during a well-child visit. Anticipatory guidance is a process whereby primary care providers discuss issues with families in anticipation of their emergence [8]. Mindell and Owens [9] highlight when to introduce and discuss issues across infancy, childhood, and adolescence in conjunction with scheduled medical well-child visits. This is an expansion on anticipatory guidance recommendations provided by the American Academy of Pediatrics (AAP) [8]. Discussions around sleep can and should take place prior to the birth of a child during the mother’s prenatal visits. During this time, age-appropriate sleep expectations, sleeping arrangements, and safe sleep practices should be discussed [9].

### 2.2. Behavioral Health Providers and Psychiatrists

The role of the behavioral health provider (i.e., a person with graduate or medical school training who has a master of social work (MSW) degree, is a licensed clinical social worker (LCSW) or licensed professional counselor (LPC), or a master’s-level psychologist, doctoral-level psychologist, or psychiatric nurse practitioner or psychiatrist) is to offer psychoeducation and empirically supported interventions to address common sleep problems. Topics that may be addressed by a behavioral health provider include, but are not limited to, behavior management for bedtime refusal behaviors; establishment and implementation of a bedtime routine; strategies to improve independent sleep onset and management of night wakings, to decrease nighttime feeds, or to decrease parasomnia concerns such as sleepwalking and sleep talking; and intervention to address circadian rhythm disorders in adolescents and teenagers. Due to the high rate of occurrence of these issues, it is advised that all behavioral health providers screen all patients for sleep difficulties. Adherence strategies to improve the use of medical interventions for sleep disorders (e.g., adherence to a continuous positive airway pressure (CPAP) machine for OSA) may also be addressed.

Behavioral health providers may provide services within an integrated primary care setting or work collaboratively with pediatricians in the community to help assess and treat behaviorally based sleep issues. Behavioral health providers in primary care practice can implement the Sleep Checkup, a clinical tool designed to prevent, identify, and manage pediatric sleep difficulties [10]. Behavioral health providers are also encouraged to coordinate with school personnel to help them understand the functional impact of sleep on behavior, academic performance, and emotional regulation. Behavioral health providers may also be involved in the assessment of attention-deficit hyperactivity disorder (ADHD), which requires getting feedback from teachers. It is estimated that 25–50% of youth with ADHD have difficulty initiating and maintaining sleep, with sleep issues occurring in two to three times more youth with ADHD than those without ADHD [9]. Behavioral health providers should screen for sleep concerns to help ensure appropriate diagnosis when comorbidities are present between behavioral issues, poor emotional regulation, anxiety, depression, and other psychiatric concerns. Considering the presence of sleep issues and their impact on emotional and behavioral regulation is crucial to prevent misdiagnosis and inappropriate medication management. Screening should include timing of sleep onset, night wakings, restless sleep, snoring, sleep regularity and duration, and daytime sleepiness [9]. Similarly, with ADHD, behavioral health providers should consider sleep issues when screening for and treating anxiety and depression. It is estimated that 40–50% of children with emotional regulation issues have comorbid sleep concerns [9]. Behavioral health providers can also offer practical recommendations to parents on managing sleep problems and work collaboratively with school systems in the US on incorporating sleep recommendations and school-based accommodations through either a 504 Plan or individualized education plans (IEPs). Periodic screening of sleep concerns should also be employed as a means of monitoring progress and the impact on daytime functioning [9].

### 2.3. Childcare Providers and School Personnel

Childcare professionals and school personnel such as teachers, nurses, social workers, and school psychologists can play an essential role in identifying daytime sleepiness and participating in intervention and monitoring of the impact of sleep issues on academic performance, emotional regulation, and daytime behavior. Among school-age youth, poor sleep has implications such as decreased academic performance, truancy, behavior problems, increased internalized symptoms consistent with anxiety and depression, memory and attention difficulties, and safety concerns [11]. Knowing the possible impact of poor sleep on daytime functioning, it is important that school personnel screen for sleep problems when assessing for learning problems and designing IEPs or school-based accommodations [12]. Without considering these factors, youth may be misdiagnosed or inappropriately classified as having ADHD or emotional disturbance (ED), or meet special education criteria for learning problems when also experiencing sleep issues.

In addition to early identification of sleep issues, strategies to help prevent and treat sleep problems can be provided by school personnel. Psychoeducation and tips for prevention and intervention of sleep problems can be incorporated into the general curriculum and in health classes as well as group lessons for parents and teachers. Wilson and colleagues [13] evaluated the impact of preschool-based sleep education targeting parents and teachers across a two-week period. Results demonstrated an increase in parents’ knowledge, attitudes, self-efficacy, and beliefs around sleep as well as increased weeknight sleep for children compared to a group who did not attend the training. Another school-based family intervention found improved sleep habits for first-grade students over a 12-month period after families received a brief consultation around sleep concerns [14]. Among older adolescents, similar improvements in sleep habits and knowledge were found following school-based sleep intervention [15,16].

Schools can also implement system-wide changes to help decrease sleep problems among children and adolescents. In one study, delaying school start time by one hour resulted in increased sleep duration, decreased attempts to “make up” sleep on the weekends, and decreased motor vehicle accidents [17]. In another study, start times were delayed by 30 min, resulting in an increase in total sleep duration and an overall decrease in daytime sleepiness among students [18]. Similar results were found in a larger longitudinal study among students in China [19]. The AAP [20] has expressed strong support for school administrations considering changing to later start times. Other recommendations include schools educating parents and students on optimal sleep durations for adolescents and teenagers, health care providers within schools serving as advisors to help schools be more aware of youth sleep needs, supporting schools that provide educational interventions for youth and parents, and educating the general community on potential risks of chronic sleep loss in adolescents.

### 2.4. Dentists and Otolaryngologists

Obstructive sleep apnea is a prevalent sleep disorder that has significant implications for pediatric patients if left untreated, including behavioral, academic, social-emotional, and medical. Within the discipline of pediatric dentistry, there has been increased attention on pediatric dentists screening for problems and referring patients for consultation with providers outside of dentistry if sleep problems are suspected. In 2016, the American Academy of Pediatric Dentistry (AAPD) adopted a policy strongly encouraging dentists to screen for possible OSA, snoring, and other sleep-disordered breathing concerns, assess for tonsillar hypertrophy, assess tongue positioning for possible obstruction, recognize that obesity can contribute to the prevalence of OSA, refer to an appropriate professional if OSA is suspected (e.g., otolaryngologist, sleep medicine physician, pulmonologist), and consider nonsurgical oral appliances only after “complete orthodontic/craniofacial assessment of the patient’s growth and development as part of a multidisciplinary approach” [21]. The AAPD’s recommendations support an interdisciplinary approach to identification and treatment of pediatric sleep concerns and can help in early detection of sleep concerns. The AAPD recommends that children first see a dentist once their first tooth erupts or by their first birthday. It is recommended that children see a dentist about every six months thereafter [22]. Thus, if a child is seeing a dentist at regular intervals, the dentist will have an important role in early identification of sleep-disordered breathing and tonsillar hypertrophy, recognizing abnormal craniofacial growth patterns, and screening for sleep-disordered breathing concerns [23]. Obstructive sleep apnea screening is also essential before recommending sedation for a pediatric patient, as children may be more susceptible to airway collapse during a procedure and recovery if they also meet the criteria for OSA [24]. The modified STOP-BANG questionnaire (snoring (S), tonsillar hypertrophy (T), obstruction (O), daytime tiredness, behavior problems or daytime irritability (P), BMI (B), age (A), Neuromuscular Disorder (N), Genetic/ Congenital Disorder (G)) for pediatrics is one screening tool that has shown potential utility among dentists in screening for OSA risk factors [25]. Aside from OSA and other sleep-disordered breathing concerns, bruxism is a highly prevalent sleep disorder among pediatric patients, which can increase sleep arousals and impact daytime functioning [26]. In fact, in a small sample of pediatric patients aged 5 to 18, there was a statistically significant difference in the arousal index with bruxism compared to age- and sex-matched controls [26].

Aside from primary care physicians and dentists, otolaryngologists are the medical professionals children are most likely to encounter. Otitis media with effusion (OME) commonly presents in children, especially between the ages of 1 and 3 years, with a prevalence of 10–30% and a cumulative incidence of 80% by age 4 [27]. Myringotomy with insertion of tympanostomy tubes is the most common operation among children in the United States [28]. The high incidence of ear infections and frequent need for tubes result in a relatively high percentage of children who are seen in otolaryngology clinics. Otolaryngologists can play a role in the screening and treatment of medically related sleep disorders. Of the nearly 500,000 pediatric tonsillectomies and adenoidectomies (T&As) performed each year, the majority are to treat sleep-disordered breathing. Otolaryngologists, who are specialists in upper airway anatomy, physiology, and surgery, are uniquely qualified to treat patients with OSA. Enlarged tonsils and adenoids are a common cause of sleep-disordered breathing. As such, surgical removal of the tonsils and adenoids is the primary treatment. In the pediatric population, resolution of OSA occurs in 82% of patients who are treated with T&A [29]. A recent study estimated that the prevalence of OSA among obese children and adolescents was as high as 60% [30]. Obstructive sleep apnea may be more severe in obese children and adolescents compared to age-matched nonobese children and adolescents [31].

## 3. Multidisciplinary Sleep Model

Infants, children, and adolescents function across a variety of systems, and poor sleep can affect more than one area of a child’s life. For example, a child who does not get enough sleep at night may experience daytime drowsiness at school, impacting academic performance, as well as irritability during after-school sports practice, impairing social relationships. This paper posits that the various professionals within these systems should screen for and intervene on sleep-related concerns to positively impact the health and well-being of all children and should work to communicate with children and families as well as across disciplines to provide coordinated care to identify, screen for, and assist in the treatment of pediatric sleep problems (Figure 1).

A medical provider such as a primary care provider or pediatrician routinely monitors a child’s sleep starting at the mother’s prenatal visits by educating the mother on expectations and establishing the sleep environment, continuing over the life of the child during well-child visits. Other medical providers such as dentists, orthodontists, and otolaryngologists can screen for possible obstructive sleep apnea or bruxism. Childcare and education professionals can help by informing caregivers of a child’s level of daytime alertness and mood. Given the significant implications of sleep on behavior and learning, it is appropriate and necessary that these professionals provide interprofessional care. Unfortunately, there is limited research and discussion on a coordinated, multidisciplinary approach to the assessment and treatment of sleep issues.

## 4. Discussion

A multidisciplinary approach to screening and prevention can help with early identification of and intervention for sleep problems. Table 1 outlines anticipatory guidance for sleep-related topics across ages that can be introduced and discussed with patients/families by the team of providers. Sleep issues may also arise at ages other than what is presented in this table. Therefore, multidisciplinary providers are encouraged to use this as a guide and consistently gather relevant information as issues arise outside of the context of regular anticipatory guidance.

Consequently, implications of poor sleep such as daytime sleepiness, academic difficulties, poor social-emotional regulation, and behavior difficulties can be mitigated. Other prospective strengths of this model include collaboration among providers, which can streamline treatment approaches and enhance coordination of care for the patient. Streamlined prevention and treatment strategies can result in more consistent and clear messages to families, increasing the likelihood of implementation. Barriers to this approach include difficulties sharing health information, as providers across systems often do not have integrated medical records unless they are within the same health agency. Without integrated health records, providers require release of information forms to be completed by families to allow coordinated communication among providers. This can be difficult to complete if it is not a regular procedure among providers. Additionally, time is a barrier to this multidisciplinary approach. True coordinated care among providers to address sleep issues requires communication via electronic health record, fax, secure email, or phone. Each mode of communication has different procedures, and using these procedures impacts a medical provider’s day differently. Often providers do not have time built into their schedule for consultation and collaboration, and often there is no additional financial compensation for the time required to execute these activities well. An additional barrier to the use of this approach is not having enough providers or having many multidisciplinary providers within a given geographic area, making it difficult to establish and potentially maintain collaboration. Geographic locations that require providers to be significant distances apart can also present a barrier, because patient follow-through for appointments may be difficult due to transportation difficulties. Integrated approaches that have a collocated or preexisting coordinated (i.e., interdisciplinary clinic) structure can help in navigating this barrier [32].

At this time, there is limited empirical evidence supporting a multidisciplinary approach for the assessment and treatment of sleep problems. While some professional associations have established best practice guidelines (e.g., pediatrics, dentistry) regarding assessment of sleep difficulties in pediatric patients, not all relevant specialties have established guidelines, and certainly no guidelines have taken into consideration coordination of care across environments. It is also important to note that specialty providers, including neurologists, endocrinologists, and geneticists, can play a role in the screening and prevention of childhood sleep problems. This review does not discuss their role, as the general pediatric population is less likely to encounter these professionals in routine care. To explore the utility of a multidisciplinary approach, empirical research studies should be employed evaluating the process as well as clinical outcomes data measuring the impact on patient care. Data examining whether a multidisciplinary approach allows patient access to services sooner may further support this model as well as attenuate the severity of symptoms. Research should also explore possible solutions to the aforementioned barriers and propose strategies to help improve the limited widespread nature of this approach.

## Figures and Tables

**Figure 1 medsci-06-00077-f001:**
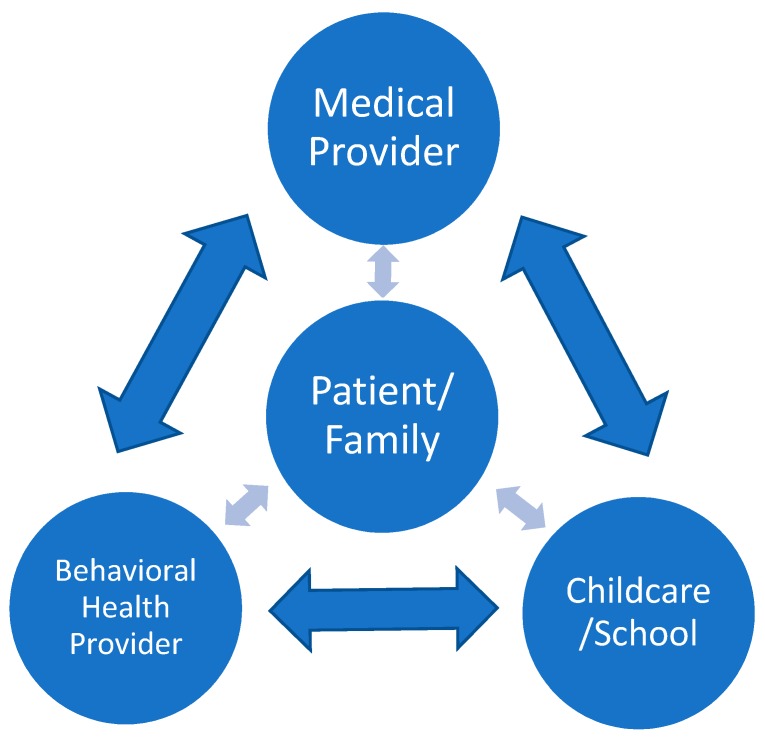
Multidisciplinary approach to the assessment and intervention of sleep problems with emphasis on communication between patients/families and providers as well as across providers.

**Table 1 medsci-06-00077-t001:** Multidisciplinary anticipatory guidance recommendations [9,10,21,22,23,24,25,26].

Sleep Anticipatory Guidance: Infants	
**Primary Care Provider**	1 to 6 Months - Encourage parents to get plenty of sleep and sleep when infant is sleeping - Help baby wake for feedings by light patting, changing the diaper, or undressing - Continue to offer feeds during the night every 3 h - Put infant to sleep on his/her back; choose a crib with slats 2 ⅜ inches apart; do not use loose, soft bedding - Put baby to sleep drowsy but awake - Pay attention to infant’s cues for sleep - Develop a schedule for naps and nighttime sleep - Infant should sleep in crib in caregiver’s room - Do not but baby in crib with a bottle - Create daily routine for naps and bedtime for baby - Choose mesh playpen with weave less than ¼ inches 7 to 12 Months - Discuss changing sleep pattern - Discuss limit setting and positive discipline - Nighttime feeds not necessary
**Behavioral Health Provider**	1 to 6 Months - Provide coping skill recommendations to caregivers to help with transition of having a newborn at home and impact on parental sleep and stress level - Help family set a consistent schedule and routine for sleep - Provide psychoeducation on sleep-onset associations - Discuss routine for feeds - Provide psychoeducation on daytime disruptive behavior management (i.e., differential attention) 7 to 12 Months - Help family gradually reduce nighttime feeds - Further discuss limit-setting techniques and positive discipline
**Childcare/School**	1 to 6 Months - Maintain regular sleep and feeding schedules - Maintain safety recommendations - Put baby to sleep drowsy but awake - Implement consistent routine for sleep - Provide feedback to caregivers on daytime sleep habits - Support independent sleep onset and reduce feedings during naps 7 to 12 Months - Provide family with feedback on helpful behavioral strategies and positive discipline techniques used at daycare - Monitor sleepiness outside of daily sleep schedule - Monitor developmental performance (i.e., cognitive, oral, and motor development)
**Dentist/Otolaryngologist**	- First visit with a dentist by the time of eruption of first tooth or first birthday - Screen for breathing concerns, oral and craniofacial abnormalities, and obstructions
**Sleep Anticipatory Guidance: Toddlers (1 to 3 Years)**	
**Primary Care Provider**	- Continue one nap per day - Follow nightly bedtime routine - Encourage quiet time such as reading, singing, and a favorite toy before bed - Maintain consistent bedtime routines and sleep times - Discuss night awakenings: parents should reassure briefly, give a preferred object (blanket or stuffed animal), and put back to bed - Do not put TV, computer, or digital device in bedroom - No bottle in bed - Use methods other than TV or digital media when tired to improve calming behavior
**Behavioral Health Provider**	- Discuss nap schedule so as to not disrupt nighttime sleep - Discuss use of transitional object for sleep and how to decrease maladaptive sleep onset associations - Discuss limit setting around electronics and digital media for sleep
**Childcare/School**	- Maintain consistent naptime earlier in the afternoon to avoid impact on nighttime sleep - Use transitional object at naptime - Continue to monitor developmental gains and recommend early intervention services or developmental assessment as indicated - Assess for sleep concerns if developmental delays appear evident
**Dentist/Otolaryngologist**	- Encourage regular dental visits (i.e., every 6 months) - Dentist discusses incorporating nightly oral hygiene habits into bedtime routine - Dentist screens for consumption of sugary and caffeinated drinks and provide education on impact of dental health and sleep - Screen for tonsillar hypertrophy, oral and craniofacial abnormalities, and nighttime breathing concerns and mouth breathing; may use pediatric-adapted screening tools such as STOP-BANG [25] - Dentist assesses for and provide psychoeducation about bruxism - Dentist discusses use of positional therapy to reduce snoring or bruxism - Otolaryngologist screens for obstructive sleep apnea (OSA)
**Sleep Anticipatory Guidance: School-Aged Children**	
**Primary Care Provider**	- Create and maintain a calm bedtime routine - Limit TV to no more than 1 h a day, no TV in bedroom - Monitor school performance and consider impact of poor sleep on tardiness, daytime behavior - Consider implementing a family media plan to balance needs of physical activity, sleep, school, and quiet time without media (www.healthychildren.org/mediauseplan) - Maintain consistent sleep routine (even on weekends) to obtain adequate sleep - Do not operate machinery, especially motor vehicles, when drowsy - Discuss maintaining a sleep routine in light of other activities, work, school, exercise, extracurricular activities, free time - Provide psychoeducation around proper use of melatonin if used
**Behavioral Health Provider**	- Help family establish a consistent bedtime routine that is not too long (e.g., bath, brush teeth, PJs, story, lights out) - Encourage daytime exercise and limit electronics use; eliminate TV and other screens at least 1 h before bed - Introduce Cognitive Behavioral Therapy (CBT) strategies for older children to help calm bedtime fears, anxiety, and mood concerns - Help family implement behavioral strategies for bedtime refusal, night awakenings, and parasomnias - Discuss daily schedule to maintain balance between school, friends, homework, and work - Discuss setting limits around driving a vehicle if sleep deprived
**School**	- Monitor drowsiness in school, report episodes of sleep during school day to caregivers - Monitor academic and behavioral performance; assess sleep difficulties when evaluating concerns - Introduce psychoeducation on sleep during class time and to parents during parent-teacher meetings and back-to-school night - Monitor tardiness, school attendance, and changes in mood or anxiety levels - Encourage regular exercise (e.g., PE classes) - Consider changing school start times - Provide psychoeducation on the impact of poor sleep on driving behavior and safety - Manage school schedules so extracurricular activities do not occur too early in the morning or too late at night
**Dentist/Otolaryngologist**	- Dentist assesses for tooth wear and screen for bruxism if wear is evident - Otolaryngologist screens for and monitors tonsillar hypertrophy and sleep-disordered breathing concerns; screens for OSA before sedating a child for oral surgery - Otolaryngologist discusses impact of obesity on breathing-related sleep disorders - Otolaryngologist discusses impact of decongestants and corticosteroids on sleep - Otolaryngologist screens for nocturnal enuresis in patients who snore - Discuss nonsurgical appliances to help correct oral abnormalities that may impact sleep-disordered breathing

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
