# Peer review of "It Takes a Village: Multidisciplinary Approach to Screening and Prevention of Pediatric Sleep Issues"

_medsci, 2018, doi:10.3390/medsci6030077_

Round 1

Reviewer 1 Report

The authors have presented a timely review of multidisciplinary nature of pediatric sleep disorders and suggested a model of collaborative, team based approach to identification and management in this population. The authors have done a remarkable job in providing a comprehensive table of anticipatory guidance across ages for each discipline as well as clear visual of the proposed model. The current limitations related to electronic medical records, time and reimbursement are presented thoughtfully.

The following suggestions are intended for authors’ review to improve the flow and expand few areas for clarity and impact.

1 – Introduction: Page 1, Line 36 and Line 43. It may be more appealing to present information on medically based sleep disorders followed by behaviorally based sleep disorders to highlight that the difference in prevalence and nature in children vs adults. It would also be prudent to use this statistic to make a case for multidisciplinary, proactive approach for sleep issues in children.

2 – Page 2. Authors may consider different title for section 2, for example, ‘Pediatric sleep across disciplines’ to better suit the descriptions of different disciplines and their role. This would be followed by ‘Multidisciplinary model’. It would be easier for readers to have the background of each discipline and their role before the multidisciplinary model is presented, potentially as a separate section or as part of discussion section.

3 – Page 3, 2.2 Behavioral health providers. The authors have thoughtfully presented the role of behavioral health in sleep. It would be of value to add a brief comment about comorbidity of sleep disorders with psychiatric disorders and the bidirectional relationship between the these two areas. This section should also include role of psychiatrists in screening and assessment of sleep issues during visits for psychiatric issues.

4 – Page 4. Line 122-123: “……misdiagnosed as having ADHD….”. May add ‘emotional disturbance’ since it is also commonly used misclassification for kids with sleep issues resulting in learning and behavioral issues.

5 – Page 5. Table 1. May consider expanding the acronym PCP, BPH in absence of any legends.

6 – Page 7. Authors may consider expanding on the implications of proposed model of care and anticipated benefits of the approaches listed in Table 1. This would also be a more appropriate place for figure describing the model.

Overall, the authors have provided a remarkable, comprehensive and practical summary. It would be an important contribution and impetus for further research.

Author Response

Point 1: Please see changes made on Page 1, lines 32-35.

Point 2: Section's 2 and 3 reordered starting on page 2. Title update as suggested- authors agree this is a more appropriate flow and title change.

Point 3: Comorbidity and bidirectional relationship between sleep and psychiatric disorders addressed on page 2, starting line 77-99.

Point 4: Thank you for this feedback. Inclusion of emotional disturbance was added to page 3 like 111.

Point 5: Table updated with expanded titles of PCP, BHP due to exclusion of legend.

Point 6: Implication expanded in discussion section, page 5 through page 8. Figure 1 moved to page 5 as suggested.

Thank you for provided feedback of this manuscript. We believe that the revisions made based upon feedback strengthen the utility of this manuscript.

Reviewer 2 Report

This is a well-written expert opinion, based on evidence, of a proposed multidisciplinary approach to sleep issues. There are two issues to consider:

The role of the behavioural health provider needs to be further described. What would be the qualifications of this person, and how would maintenance of competence be assessed. Are you considering 'sleep consultants' the same as behavioral health providers?  If so, what is the minimal training and certification, and what type of children would the BHP be able to provide counselling about sleep for? 

This approach may only apply to the USA but this is not stated. For example, on line 109, the authors discuss 504-plans. This would not be familiar to all readers. Also, the authors should describe the availability and cost for the BHP, after expanding on their qualifications as above.

Author Response

Point 1: Behavioral health provider definition and recommended training expanded upon on page 2, lines 64-68 per feedback provided by this reviewer. Authors are not suggesting sleep consultants are the same as behavioral health providers for the context of this manuscript.

Point 2: Discussion around use of IEP and 504-plans clarified to indicate system in place within the U.S. school system. Please see page 3, line 95-99.

Reviewer 3 Report

The topic of the article is very intresting and useful. Multidisciplinary approch to assesment  of pediatric sleep issues is very important.

In my opinion,  the title  of the article it would be better to remove  "treatment" as there are non indications for the treatment of all sleep disorders.

I would add the very important role of multisciplinary  approch in the management of sleep disorders in children  with genetic syndromes, neromuscolar diseases.

157  I would also add:  endocrinologist

Author Response

Point 1: Manuscript title changed as follows to indicate less emphasis on treatment. "It takes a village: Multidisciplinary approach to screening and prevention of pediatric sleep issues." Authors agree that the emphasis of this manuscript is more on screening and prevention, rather than recommendations for treatment.

Point 2: Role of other specialty providers, such as Neurologists, Endocrinologists, and Geneticists can play an important role for the screening and prevention of sleep issues among pediatric populations. Please see page 8, lines 242- 246.

Round 2

Reviewer 2 Report

This revisions to this manuscript are well written.